# Sesaminol Inhibits Adipogenesis by Suppressing Mitotic Clonal Expansion and Activating the Nrf2-ARE Pathway

**DOI:** 10.3390/nu17203242

**Published:** 2025-10-15

**Authors:** Saki Nakamatsu, Miki Nakata, Toshio Norikura, Yutaro Sasaki, Isao Matsui-Yuasa, Ayano Omura, Kunio Kiyomoto, Akiko Kojima-Yuasa

**Affiliations:** 1Department of Nutrition, Graduate School of Human Life and Ecology, Osaka Metropolitan University, Osaka 538-8525, Japan; nakamatsu.saki299@gmail.com (S.N.); mamemiki710@gmail.com (M.N.); sy23283r@st.omu.ac.jp (Y.S.); yuasa-i@hotmail.co.jp (I.M.-Y.); 2Department of Nutrition, Aomori University of Health and Welfare, Aomori 030-8505, Japan; t_norikura@ms.auhw.ac.jp; 3Kiyomoto Co., Ltd., Miyazaki 889-0595, Japan; oomura-aya@kiyomoto.co.jp (A.O.);

**Keywords:** sesaminol, adipogenesis inhibition, mitotic clonal expansion (MCE), Nrf2-ARE pathway, reactive oxygen species (ROS), C/EBPβ centromere

## Abstract

Background: As a key contributor to metabolic disorders, obesity is recognized as a critical global health challenge. Adipocyte differentiation depends on the mitotic clonal expansion (MCE) phase, which is controlled by oxidative balance and transcription factors like C/EBPβ. Sesaminol, a lignan derived from *Sesamum indicum*, has potent antioxidant properties. This study aimed to investigate whether sesaminol suppresses adipogenesis by modulating ROS signaling, MCE, and the Nrf2-ARE pathway. Methods: In the early period of adipogenic induction, 3T3-L1 preadipocytes received treatment with sesaminol. Adipogenic development was evaluated through Oil Red O staining together with the assay of GPDH activity. Assays of cell proliferation and expression of cell cycle-related proteins, along with ROS measurement, qRT-PCR, Western blotting, and immunofluorescence, were performed to evaluate the effects on oxidative stress, transcriptional regulation, and AMPK-Nrf2 signaling. Results: Sesaminol significantly inhibited lipid accumulation and GPDH activity without cytotoxicity. It suppressed MCE by inhibiting DNA synthesis and reducing the expression of cyclin E1/E2 and CDK2. Sesaminol decreased C/EBPβ expression and its nuclear localization, resulting in lower levels of C/EBPα and PPARγ. It also reduced intracellular ROS, promoted nuclear translocation of Nrf2, and upregulated antioxidant genes HO-1 and GCLC. AMPK phosphorylation was concurrently enhanced. Conclusions: Sesaminol inhibits early adipogenesis by suppressing ROS-mediated MCE and activating the AMPK-Nrf2-ARE signaling pathway, leading to downregulation of key adipogenic transcription factors. The present study supports the potential of sesaminol as an effective strategy for obesity prevention.

## 1. Introduction

As obesity is strongly correlated with metabolic disorders—such as type 2 diabetes, hypertension, and lipid abnormalities—it has emerged as a major global health issue. These metabolic disturbances promote the development of metabolic syndrome and are risk factors for cardiovascular disease, chronic inflammatory states, and neurodegenerative diseases. Despite the availability of pharmacological interventions, effective and safe long-term treatments for obesity remain limited, necessitating the exploration of novel therapeutic targets and compounds.

Adipogenesis, defined as the maturation of preadipocytes into adipocytes, constitutes a fundamental mechanism contributing to obesity. The differentiation of preadipocytes is a multistage process, beginning with mitotic clonal expansion (MCE) and continuing through early and terminal differentiation, with MCE being an essential phase that governs preadipocyte proliferation and subsequent differentiation [1,2]. Targeting MCE has been proposed as a potential anti-obesity strategy, as its disruption can prevent adipocyte formation at an early stage [3]. However, effective molecular regulators capable of safely modulating this process remain largely unexplored.

During MCE, preadipocytes transition from the quiescent G0/G1 phase to the G2 phase, undergoing multiple rounds of cell division [3,4,5]. This process is controlled by transcription factors, including CCAAT/enhancer-binding protein beta (C/EBPβ), which is crucial for the initiation of adipocyte differentiation [6]. Signaling mediated by reactive oxygen species (ROS) has also been shown to influence C/EBPβ activity and other adipogenic transcription factors, suggesting that oxidative stress modulation could be a promising strategy for regulating adipogenesis [7].

Sesaminol, a lignan derived from sesame seeds (*Sesamum indicum* L.), is known for its potent antioxidant properties and anti-inflammatory effects [8,9]. Recent studies have demonstrated its neuroprotective effects via oxidative stress modulation, particularly in the context of Parkinson’s disease, highlighting its ability to regulate redox-sensitive signaling pathways [10]. Given the established role of ROS in adipogenesis, we hypothesize that sesaminol may exert anti-adipogenic effects by modulating oxidative stress and cell cycle progression during the MCE phase.

This study aims to investigate whether sesaminol can inhibit adipogenesis by targeting key molecular events in the MCE phase, including ROS-mediated activation of C/EBPβ and cell cycle progression, thereby identifying its potential as a novel pharmacological candidate for obesity management.

## 2. Materials and Methods

### 2.1. Preparation of Sesaminol from Defatted Sesame Cake

Sesaminol is abundantly present in sesame oil cake, mostly in the form of sesaminol triglucoside (STG). However, since STG exhibits little antioxidant activity in vitro, its sugar moieties need to be hydrolyzed. Due to its branched sugar configuration and the steric hindrance of the aglycone, STG is largely resistant to enzymatic hydrolysis by typical β-glucosidases. Due to its branched sugar configuration and the steric hindrance of the aglycone, STG is largely resistant to enzymatic hydrolysis by typical β-glucosidases. In a search for microorganisms capable of producing enzymes that can hydrolyze STG, a bacterial strain, *Paenibacillus* sp. KB0549, was isolated from sesame oil cake [11]. Sesaminol was purified from defatted sesame cake following extraction and isolation procedures. *Paenibacillus* sp. KB0549 was cultivated in a liquid medium of 1% tryptone, 0.5% yeast extract, and 0.89% NaCl, using a hot-water extract of the defatted sesame cake as the base. After cultivation, the culture broth was mixed with heat-sterilized sesame cake and subjected to solid-state fermentation at 37 °C for 6 days. During fermentation, the mixture was periodically stirred and aerated to maintain optimal conditions.

Following fermentation, the sesame cake was dried and extracted with 95% ethanol at 50 °C under continuous stirring. The ethanol extract was then filtered and concentrated under reduced pressure. To obtain a highly concentrated sesaminol solution, 99.5% ethanol was added to the concentrate, followed by evaporation as described previously [11]. Quantification of sesaminol was performed via high-performance liquid chromatography (HPLC) (Appendix A). The extract was freeze-dried and subsequently dissolved in dimethyl sulfoxide (DMSO) for further use.

### 2.2. Cell Culture

3T3-L1 preadipocytes (JCRB9014), obtained from the Japanese Cancer Research Resources Bank (Ibaraki, Japan), were maintained in Dulbecco’s modified Eagle’s medium (DMEM; Shimadzu Diagnostics Corporation, Tokyo, Japan) supplemented with 10% fetal bovine serum (FBS; NICHIREI BIOSCIENCES INC., Tokyo, Japan). For adipogenic differentiation, cells were plated at a density of 1 × 10^5^ cells per dish. When the culture reached full confluence, differentiation was initiated two days later (designated as day 0) by treating the cells with DMEM containing 10% FBS, 0.25 μM dexamethasone, 0.5 mM 3-isobutyl-1-methylxanthine, and 0.2 μM insulin (DMI) for 48 h. Following this induction, cells were cultured in DMEM with 10% FBS and 0.2 μM insulin for an additional 48 h, and then in DMEM with 10% FBS alone for four more days to achieve complete differentiation.

Sesaminol was prepared in DMSO and added to the culture medium, keeping the final DMSO concentration below 0.5%. Control cells received the same medium containing DMSO only.

### 2.3. Cell Viability Assay (Neutral Red Method)

The Neutral Red uptake assay was used to evaluate cell viability by detecting the incorporation of the dye into lysosomes of viable cells [12]. Following a 24-h incubation at 37 °C, the medium was discarded, and Neutral Red dye (FUJIFILM Wako Pure Chemical Corporation, Osaka, Japan) was applied at 50 µg/mL. The cells were then incubated for an additional 2 h at 37 °C. Following incubation, the cells were gently washed with 2 mL of a 1% formaldehyde/1% Cacl_2_ solution to remove excess dye. After adding 1 mL of an acetic acid/50% ethanol solution to each well to release the dye, the cells were allowed to sit at room temperature for 30 min. The optical density of the extracted solution was subsequently measured at 540 nm with a JASCO V-730 BIO spectrophotometer (JASCO Corporation, Tokyo, Japan).

### 2.4. Measurement of Triacylglycerol (TG) Accumulation

Adipocyte differentiation was determined by Oil Red O staining [13]. After aspirating the culture medium, the cells were washed with Ca^2+^/Mg^2+^-free PBS, fixed in 60% ethanol, and incubated with 1 mL of Oil Red O solution for 2 h at room temperature. Following removal of excess dye with 50% ethanol and two washes with ultrapure water, lipids were extracted using 2-propanol (1 mL), and the absorbance of the extract was measured at 520 nm with a JASCO V-730 BIO spectrophotometer.

### 2.5. Analysis of Glycerol-3-Phosphate Dehydrogenase (GPDH) Activity

The adipocytes were washed twice with 1 mL of PBS (-) and harvested into 350 μL of triethanolamine/EDTA buffer. Cells were then lysed via sonication using a BIO RUPTOR (Cosmo Bio Co., Ltd., Tokyo, Japan). Following lysis, the samples were centrifuged at 13,000 rpm for 5 min at 4 °C, and the resulting supernatant was used for the enzyme assay. GPDH activity was measured by monitoring the rate of NADH oxidation over a period of 3 min, utilizing an extinction coefficient of 6.22 mM^−1^ cm^−1^ [14]. The enzyme activity was normalized to the control, which was set at 100%.

### 2.6. Quantitative Reverse Transcription-Polymerase Chain Reaction (qRT-PCR)

RNA was extracted from 3T3-L1 preadipocytes using the High Pure RNA Isolation Kit (Roche, Basel, Switzerland), and its yield and quality were subsequently evaluated with an Agilent 2100 Bioanalyzer (Agilent Technologies, Santa Clara, CA, USA). Complementary DNA (cDNA) was synthesized from total RNA using the PrimeScript RT Reagent Kit (TaKaRa Bio Inc., Shiga, Japan). Quantitative real-time PCR (qRT-PCR) was carried out on a StepOnePlus Real-Time PCR System (Thermo Fisher Scientific, Waltham, MA, USA) with TB Green Premix Ex Taq II (TaKaRa Bio Inc.) as the detection reagent. The amplification program consisted of an initial denaturation step at 95 °C for 30 s, followed by 40 cycles of 95 °C for 5 s and 60 °C for 30 s. Primer sequences used in this study are provided in Table 1. Gene expression levels were normalized against β-actin, and relative quantification was calculated with the ΔΔCT method using StepOne software version 2.2.2 (Thermo Fisher Scientific).

### 2.7. Western Blotting

Western blot analysis was conducted to assess protein expression in 3T3-L1 adipocytes. After rinsing the cells twice with 1 mL of PBS (-), they were collected in 150 μL of RIPA buffer using a cell scraper. The obtained lysates were subjected to sonication, followed by centrifugation at 15,000 rpm for 10 min at 4 °C. The supernatant was mixed with sample buffer consisting of 1 M Tris-HCl (pH 6.8), 15% SDS, 30% glycerol, 20% 2-mercaptoethanol, and 1% bromophenol blue, heated at 90 °C for 5 min, and stored at –80 °C until further analysis. Protein concentrations were determined using the Pierce™ BCA Protein Assay Kit (Thermo Fisher Scientific). A total of 20 μg of protein was resolved on 10% SDS-PAGE gels and subsequently transferred onto polyvinylidene difluoride (PVDF) membranes (Merck Millipore, Burlington, MA, USA). Membranes were blocked with 3% bovine serum albumin for 1 h at room temperature before antibody incubation. Primary antibodies were applied at the following dilutions: anti-C/EBPβ (H-7), anti-C/EBPα (14AA), and anti-PPARγ (E-8) (Santa Cruz Biotechnology, Dallas, TX, USA) at 1:2000; anti-AMPKα, phospho-AMPKα (Thr172) (40H9), and GAPDH (D16H11)XP rabbit mAb (Cell Signaling Technology, Danvers, MA, USA), or anti-β-actin (Dako Denmark A/S, Glostrup, Denmark) at 1:5000 as loading controls. After overnight incubation with primary antibodies at 4 °C, membranes were treated with secondary antibodies (goat anti-mouse or goat anti-rabbit, Dako Denmark A/S) diluted 1:3000. Protein signals were visualized using EZ West Lumi One (ATTO, Tokyo, Japan) and imaged with the AE-9300 EZ-Capture MG system (ATTO). Band intensities were quantified with CS Analyzer software v3.0 (ATTO).

### 2.8. Measurement of Intracellular ROS Production

Intracellular reactive oxygen species (ROS) generation was assessed using 2′,7′-dichlorodihydrofluorescein diacetate (DCFH-DA), a probe with relative specificity for hydrogen peroxide. Cells were exposed to 2.4 mM DCFH-DA (5 μL) during the last 30 min of treatment and subsequently rinsed twice with PBS. Fluorescence signals were captured with an FSX100 Bioimaging Navigator system (Olympus Corporation, Tokyo, Japan), and ROS levels were quantified based on fluorescence intensity.

### 2.9. Measurement of Nuclear Translocation of Nrf2

3T3-L1 cells were seeded in Lab-Tek Chamber Slides and cultured in medium containing MDI, with or without the addition of 30 μg/mL sesaminol, to induce differentiation. At 4, 16, and 24 h after induction, the cells were washed three times with 0.5 mL PBS and then permeabilized with 0.1% Triton-X (FUJIFILM Wako Pure Chemical Corporation). To reduce nonspecific antibody binding, the cells were blocked with two drops of Protein Block Serum-free for 30 min. Slides were subsequently incubated overnight at 4 °C with an anti-Nrf2 antibody (Santa Cruz Biotechnology, Inc.). After washing with PBS, the cells were stained with Alexa Fluor 488-conjugated goat anti-rabbit IgG (Life Technologies, Carlsbad, CA, USA) for 1 h. Nuclei were counterstained with 4′,6-diamidino-2-phenylindole dihydrochloride (DAPI, FUJIFILM Wako Pure Chemical Corporation). Nuclear localization of Nrf2 was then assessed using an all-in-one fluorescence microscope (BZ-X800, KEYENCE Corporation, Osaka, Japan).

### 2.10. Immunofluorescence Microscopy

3T3-L1 cells were cultured for 4, 16, and 24 h following the induction of differentiation with MDI. The culture dishes were washed twice with 0.5 mL of PBS. Fixation was performed by immersing the dishes in methanol (−20 °C) for 5 min. After fixation, the dishes were rinsed with PBS and then treated with 200 μL of PBS containing 0.1% Triton X-100 for 5 min to permeabilize the cells. Following another PBS wash, the dishes were incubated with Protein Block Serum-Free solution for 30 min. Cells were washed with PBS after blocking and then treated with the primary antibody (anti-C/EBPβ) overnight at 4 °C. The following day, the dishes were washed with PBS and incubated with a FITC-conjugated secondary antibody for 1 h. After a final PBS wash, cover glasses were mounted on the dishes, and the samples were air-dried in the dark. Fluorescence microscopy was used to observe the samples.

### 2.11. DNA Content Measurement

The DNA content of 3T3-L1 cells was determined according to the method of Burton [15]. Briefly, cells were collected by centrifugation at 500× *g* for 5 min and disrupted by three freeze–thaw cycles in 300 μL of 0.4 N perchloric acid. The lysates were centrifuged at 17,400× *g* for 20 min, and the pellets were hydrolyzed in 600 μL of 0.4 N perchloric acid at 70 °C. After a further centrifugation step at 17,400× *g* for 15 min, the supernatant was used for DNA quantification. For each sample (0.4 mL), 0.1 mL of distilled water was added together with 1 mL of DNA reagent, prepared by dissolving 1.5 g of diphenylamine in 100 mL acetic acid and 1.5 mL H_2_SO_4_, and mixing this solution with acetaldehyde (22.6 μL in 1 mL distilled water) at a ratio of 200:1. The reaction mixture was incubated at 36 °C for 16 h, and absorbance was recorded at 600 nm using a spectrophotometer.

### 2.12. Statistical Analysis

Comparisons among multiple groups were conducted using one-way ANOVA followed by Tukey–Kramer post hoc tests using Statcel-4 (OMS Inc., Tokorozawa, Japan). Significance was assessed at the 5% or 1% risk level, with data presented as mean ± SD.

## 3. Results

### 3.1. Effect of Sesaminol on the Viability of 3T3-L1 Preadipocytes

Cytotoxicity of sesaminol in 3T3-L1 preadipocytes was determined by the neutral red uptake assay following 24 h exposure to concentrations between 0 and 100 µg/mL (Figure 1A). No significant impact on cell viability was observed at concentrations below 40 µg/mL. Therefore, sesaminol concentrations below 40 µg/mL were selected for further experimentation.

### 3.2. Effect of Sesaminol on Lipid Accumulation in 3T3-L1 Preadipocytes

Following the viability assessment, the effect of sesaminol on intracellular TG accumulation was investigated during 3T3-L1 differentiation using Oil Red O staining. On day 8, the control group showed a marked increase in lipid accumulation post-differentiation, while sesaminol-treated cells exhibited a dose-dependent decrease in lipid content. Red-stained granules, representing intracellular lipids, were quantified spectrophotometrically (Figure 1B), indicating a dose-dependent reduction in TG levels by sesaminol.

### 3.3. Effect of Sesaminol on GPDH Activity in 3T3-L1 Preadipocytes

To further confirm the impact of sesaminol on lipid metabolism, we measured the activity of GPDH, a key enzyme in TG biosynthesis. Sesaminol treatment for 8 days resulted in a dose-dependent decrease in GPDH activity, which paralleled the reduction in intracellular TG content (Figure 1C), supporting its role in attenuating adipogenesis. For the following experiments, 30 μg/mL sesaminol was selected, as this concentration showed the most marked effects on TG accumulation and GPDH activity. For the following experiments, 30 μg/mL sesaminol was selected, as this concentration showed the most marked effects on triglyceride accumulation and GPDH activity.

### 3.4. Effect of Sesaminol at Different Stages of Adipogenesis

We sought to define the stage of 3T3-L1 cell differentiation during which sesaminol interferes with adipogenic progression. Sesaminol was administered at different points across the 8-day differentiation period. Oil Red O staining on day 8 revealed that sesaminol significantly suppressed TG accumulation, particularly during the first two days, coinciding with cell cycle arrest and mitotic clonal expansion (MCE) (Figure 2B).

### 3.5. Effect of Sesaminol on DNA Synthesis in 3T3-L1 Adipocytes

During MCE, preadipocytes undergo multiple rounds of DNA replication. DNA content analysis, 48 h post-induction, showed a marked increase in the control group, whereas sesaminol-treated cells exhibited no such rise (Figure 3A), suggesting inhibition of DNA synthesis. Furthermore, the expression levels of cyclin E1, cyclin E2, and cyclin-dependent kinase 2 (CDK2), key regulators of the G1-to-S phase transition, were assessed. Sesaminol treatment resulted in reduced expression of these regulators (Figure 3B–D). These findings suggest that sesaminol impedes MCE, possibly by inhibiting early phases of differentiation through the suppression of cell cycle progression.

### 3.6. Effect of Sesaminol on Key Transcription Factors in Adipocyte Differentiation

We next investigated how sesaminol influences the expression of pivotal transcriptional regulators involved in adipocyte differentiation. While C/EBPβ mRNA levels remained unchanged at 24 h after induction, its protein abundance was markedly suppressed by sesaminol (Figure 4A,B). Furthermore, sesaminol treatment led to a significant decrease in both mRNA and protein expression of C/EBPα and peroxisome proliferator-activated receptor γ (PPARγ), two downstream transcription factors regulated by C/EBPβ (Figure 4C–F).

### 3.7. Effect of Sesaminol on C/EBPβ Localization to the Centromere

C/EBPβ facilitates adipocyte differentiation by localizing to the centromere during MCE, where it promotes the expression of C/EBPα and PPARγ [16]. The centromeric localization of C/EBPβ in cells was assessed using immunofluorescence analysis. At 4 h following differentiation induction, C/EBPβ was observed in the nucleus in the control group; however, it was diffusely distributed within the nucleus and was slightly localized to the centromere (Figure 5A). At 16 h after the induction of differentiation, punctate green fluorescence was observed, indicating centromeric localization (Figure 5B). Furthermore, this localization persisted until 24 h after the induction of differentiation (Figure 5C). In contrast, in the sesaminol group, centromeric localization was suppressed at all examined time points (4, 16, and 24 h after the induction of differentiation) (Figure 5A–C).

### 3.8. Effect of Sesaminol on Intracellular ROS Production in 3T3-L1 Preadipocytes

ROS production typically increases during adipogenesis and plays a role in the activation of C/EBPβ. To examine the involvement of sesaminol in the regulation of ROS levels. To assess the involvement of sesaminol in the regulation of ROS levels during the early stages of 3T3-L1 preadipocyte differentiation, intracellular ROS was measured at 4, 18, and 26 h after induction using the DCFH-DA assay. In the control group, ROS levels increased significantly over time, whereas sesaminol treatment significantly suppressed ROS production at all time points (Figure 6A,B), suggesting its potential antioxidative role in adipogenesis. Previous research has shown that ROS levels are increased in the adipose tissue of obese mice, along with upregulated expression of NADPH oxidase on the plasma membrane [17]. To further investigate the involvement of NADPH oxidase in ROS production during early adipocyte differentiation, we assessed ROS levels in the presence of diphenyleneiodonium chloride (DPI), a specific inhibitor of NADPH oxidase. DPI treatment resulted in a marked reduction in ROS levels, indicating that NADPH oxidase contributes to ROS generation in the early stages of 3T3-L1 preadipocyte differentiation (Figure 7).

### 3.9. Effect of Sesaminol on Antioxidant Gene Expression

To elucidate the mechanism by which sesaminol suppresses ROS, we analyzed the expression of several antioxidant enzymes. Sesaminol significantly upregulated the mRNA levels of *heme oxygenase 1 (HO-1)* and *glutamate–cysteine ligase catalytic subunit (GCLC)*, key components of the cellular antioxidant defense system (Figure 8A,B). These findings suggest sesaminol selectively modulates antioxidant gene expression to enhance ROS detoxification.

### 3.10. Effect of Sesaminol on Nrf2 Nuclear Translocation

Nrf2 is a key regulator of the antioxidant response. Immunofluorescence staining at 24 h post-differentiation, the amount of Nrf2 in sesaminol-treated nuclei was significantly increased compared to control nuclei. However, the addition of compound C (CC) to sesaminol significantly reduced the increased Nrf2 levels (Figure 9A,B). The enhancement of Nrf2 nuclear translocation by sesaminol was also supported by the graph of nuclear/whole cell ratio (Figure 9C). On the other hand, the addition of CC to both control and sesaminol-treated whole cells significantly reduced Nrf2 levels. These findings suggest that AMPK is involved not only in the nuclear translocation of Nrf2 but also in its synthesis.

### 3.11. Effect of Sesaminol on AMPK Phosphorylation

Western blot analysis also revealed that sesaminol significantly increased the protein level of active phosphorylated AMPK compared to the untreated control (Figure 10A) while it decreased the protein level of total AMPK (Figure 10B). These findings suggest that sesaminol activates Nrf2 via AMPK phosphorylation, promoting an antioxidant response.

## 4. Discussion

In this study, we demonstrated that sesaminol strongly inhibits adipogenesis in 3T3-L1 cells through coordinated actions on lipid metabolism, cell cycle regulation, transcription factor modulation, and oxidative stress control.

Cytotoxicity assays confirmed that sesaminol concentrations below 40 μg/mL did not impair cell viability, validating the conditions used for subsequent experiments. Within this non-toxic range, sesaminol significantly reduced intracellular TG accumulation in a dose-dependent manner. This reduction was accompanied by decreased activity of GPDH, a key enzyme in TG biosynthesis.

Stage-specific analyses revealed that sesaminol exerted its strongest inhibitory effect during the first two days of differentiation, coinciding with mitotic clonal expansion (MCE). DNA content analysis showed that control cells underwent DNA synthesis during MCE, whereas this increase was absent in sesaminol-treated cells. Consistently, sesaminol downregulated cyclin E1, cyclin E2, and CDK2, supporting the notion that cell cycle arrest underlies impaired MCE and blocked adipogenic commitment.

At the transcriptional level, sesaminol markedly reduced the expression of C/EBPα and PPARγ, which are key transcriptional regulators driving adipocyte differentiation. Although *C*/*EBPβ* mRNA expression was unaffected, its protein abundance was strongly decreased, suggesting post-translational regulation [18]. Adipocyte differentiation is normally initiated by the early expression of C/EBPβ and C/EBPδ, which peak within two days after induction [19]. Previous studies have shown that C/EBPβ knockdown causes cell cycle arrest and suppresses adipogenesis [20,21], while C/EBPβ and C/EBPδ induce C/EBPα and PPARγ through C/EBP binding elements [4,22,23]. Once expressed, C/EBPα and PPARγ reinforce each other’s transcription and activate adipogenic gene expression, thereby enhancing insulin sensitivity [24]. Consistent with this cascade, our immunofluorescence analysis revealed that sesaminol disrupted the centromeric localization of C/EBPβ during early differentiation, an essential step for downstream transcriptional activation [16,25]. These findings indicate that sesaminol impairs both the stability and functional activation of C/EBPβ, thereby attenuating the transcriptional network required for adipogenesis.

In line with this notion, C/EBPβ plays a pivotal role during the early phase of adipocyte differentiation, where its expression and stability critically determine the induction of C/EBPα and PPARγ [26,27]. Increasing evidence indicates that C/EBPβ activity is tightly controlled by post-translational modifications (PTMs). Phosphorylation acts as a primary switch, modulating DNA-binding activity and often serving as a prerequisite for ubiquitination, which targets C/EBPβ for proteasome-dependent degradation [28,29]. Thus, phosphorylation–ubiquitination–degradation cascades represent a central mechanism regulating its half-life. In parallel, acetylation enhances transcriptional activity [30], methylation alters DNA binding and specificity [31], and SUMOylation generally represses C/EBPβ-driven transcription [32]. These modifications act in a combinatorial and context-dependent manner, integrating signaling inputs from the adipogenic milieu. Collectively, PTM-mediated regulation of C/EBPβ functions as a molecular rheostat that balances differentiation versus repression, ultimately shaping adipocyte fate. Further investigation is needed to clarify how distinct PTM patterns coordinate adipogenesis under physiological and pathological conditions.

Moreover, sesaminol upregulated HO-1 and GCLC, both key enzymes for ROS detoxification that have been implicated in the inhibition of adipogenesis [33,34]. This selective regulation suggests that sesaminol activates specific antioxidant pathways that effectively reduce intracellular ROS, thereby attenuating pro-adipogenic signaling. Supporting this view, sesaminol promoted nuclear translocation of Nrf2, a central regulator of redox homeostasis [35,36], and enhanced phosphorylation of AMPK, a known upstream activator of Nrf2 [37,38]. Recent studies further show that sesamolin increases Nrf2 protein levels (not mRNA) and upregulates Nrf2 target genes such as HO-1 and NQO1 via Keap1-mediated signaling [39], and sesaminol also enhances mitochondrial activity and suppresses ROS production in hepatocytes [40]. These findings are in agreement with previous reports that sesamin and sesamolin inhibit adipogenesis in 3T3-L1 cells by downregulating PPARγ protein expression [41]. In contrast, the significance of our study lies in elucidating the molecular background by which adipogenesis is inhibited through antioxidant effects. Specifically, sesaminol was found to suppress adipocyte differentiation through multiple coordinated mechanisms: (1) inhibition of mitotic clonal expansion (MCE) via cell cycle arrest, (2) suppression of C/EBPβ activation and the downstream transcriptional cascade, and (3) inhibition of ROS signaling through activation of the AMPK–Nrf2-dependent antioxidant pathway [42,43,44].

These convergent effects ultimately led to reduced lipid accumulation and downregulation of adipogenic gene expression, thereby highlighting the potential of sesaminol as a dietary or pharmacological candidate for the prevention of obesity and related metabolic disorders. Generally, bioactive compounds intended for pharmacological application are expected to be effective at very low concentrations. However, the effective concentration of sesaminol in the present study was 30 μg/mL, which is relatively high. Similarly, in previous anti-obesity studies using 3T3-L1 cells, sesamin was applied at concentrations of 40 μM (14.17 μg/mL) to 80 μM (28.35 μg/mL), and sesamolin at 40 μM (14.81 μg/mL) to 80 μM (29.63 μg/mL) [44]. In addition, in the study by Kim et al., which demonstrated suppression of adipocyte differentiation via Nrf2 activation by sesamolin, the concentrations ranged from 25 μM (9.26 μg/mL) to 100 μM (37.04 μg/mL) [39]. By contrast, in our in vitro Parkinson’s disease model using SH-SY5Y cells, sesaminol exerted a neuroprotective effect at a concentration of 2 μg/mL [10]. Ruankham et al. also reported that 1 μM of sesamin and sesamol remarkably reduced SH-SY5Y cell death induced by 400 μM H_2_O_2_. Collectively, these findings suggest that adipocytes exhibit relatively low sensitivity to sesame lignans.

A limitation of this study is that all experiments were conducted using the 3T3-L1 cell line, an in vitro model of adipogenesis. Although this system is widely used to investigate the molecular mechanisms underlying adipocyte differentiation, it does not fully recapitulate the systemic, hormonal, and metabolic interactions that occur in vivo. Therefore, future studies employing animal models as well as hepatocytes or primary adipocytes will be essential to validate and extend the translational potential of sesaminol as an anti-obesity agent.

Furthermore, it is conceivable that the anti-obesity effects of sesaminol could be achieved at lower concentrations when combined with other compounds possessing anti-adipogenic activities. Thus, sesaminol represents a promising candidate for further investigation of such additive or synergistic effects.

## 5. Conclusions

Our study uncovers a novel anti-adipogenic mechanism of sesaminol, characterized by inhibition of early adipocyte differentiation events. Specifically, sesaminol suppresses mitotic clonal expansion (MCE), downregulates key cell cycle regulators, and disrupts ROS-dependent activation of C/EBPβ (Figure 1). In parallel, it reinforces cellular redox homeostasis through stimulation of the Nrf2–ARE signaling pathway and selective induction of genes such as HO-1 and GCLC, ultimately mitigating oxidative stress and dampening adipogenic transcriptional activity. Further investigation is necessary to find promising candidates for the prevention of obesity and other oxidative stress-related metabolic disorders by sesaminol.

## Data Availability

The original contributions presented in the study are included in the article; further inquiries can be directed to the corresponding author.

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
