# Peer review of "Sesaminol Inhibits Adipogenesis by Suppressing Mitotic Clonal Expansion and Activating the Nrf2-ARE Pathway"

_nutrients, 2025, doi:10.3390/nu17203242_

Round 1

Reviewer 1 Report

Comments and Suggestions for Authors

Please improve the description of the statistical analysis in figure captions, through the inclusion of the ANOVA P values.

In statistical analysis paragraph, please include details about the software used for the evaluations.

In figure capions please include the dose of sesaminol.

Reviewer 2 Report

Comments and Suggestions for Authors

This manuscript by Nakamatsu and colleagues explores the anti-adipogenic properties of sesaminol in 3T3-L1 cells, focusing on its ability to suppress mitotic clonal expansion (MCE) and to activate the AMPK–Nrf2 pathway. The topic is certainly relevant to the readership of Nutrients, as it connects a dietary lignan with molecular mechanisms involved in adipogenesis and oxidative stress. The study also makes use of a broad range of experimental techniques, from viability and lipid accumulation assays to qRT-PCR, Western blotting, ROS detection, and immunofluorescence. Taken together, the results indicate that sesaminol can inhibit early adipogenesis and promote antioxidant responses.

That said, there are several important issues that weaken the manuscript in its current form.

First, the novelty of the findings is not sufficiently established. The authors themselves cite recent papers showing similar effects of other sesame lignans, such as sesamin and sesamolin, on adipogenesis and Nrf2 activation. What is new here beyond confirming that sesaminol behaves in a comparable way? The manuscript needs to make a much clearer case for its unique contribution.

Second, all of the experiments are restricted to 3T3-L1 preadipocytes. While this is a well-accepted model, the lack of validation in primary adipocytes or human-derived cells limits the translational significance of the work. At a minimum, this limitation should be clearly acknowledged, and the claims regarding therapeutic potential need to be toned down.

Another concern is the concentrations of sesaminol used. Inhibitory effects are reported at 20–30 µg/mL, which is a relatively high range and may not be physiologically achievable in vivo, given the limited bioavailability of lignans. The authors should discuss this point in greater depth and consider the relevance of their findings to realistic nutritional or pharmacological contexts.

On the mechanistic side, the study remains largely correlative. The reduction of C/EBPβ protein but not mRNA suggests post-translational regulation, but no experiments were conducted to explore this further. Likewise, although AMPK phosphorylation and Nrf2 translocation are shown, these data stop short of proving causality. Knockdown experiments or more rigorous inhibitor studies would be needed to substantiate the proposed pathway.

There are also issues with data presentation. Some figures, such as the C/EBPβ immunofluorescence images, are shown without quantitative analysis. Western blots are not consistently backed up by densitometry from multiple replicates, and in some cases the sample size is as low as three. These are acceptable for preliminary results but not for a paper aiming at high impact.

The conflict of interest statement should also be expanded. Two authors are affiliated with a company that supplied sesaminol, but the disclosure is vague. The role of this company in extraction, study design, and interpretation must be clarified to ensure transparency. Moreover, the manuscript would benefit from language polishing. There are typographical errors and awkward phrasing throughout, and the schematic at the end is oversimplified. It should be redrawn to provide a more integrative view of the proposed mechanism.

In summary, while the study is interesting and contributes additional evidence on the biological effects of sesaminol, its novelty, mechanistic depth, and translational relevance are limited. I recommend major revision, with particular attention to clarifying what is new, strengthening mechanistic evidence, addressing the physiological plausibility of the findings, and improving data presentation and transparency.

Reviewer 3 Report

Comments and Suggestions for Authors

Saki Nakamatsu et al. report an interesting study entitled “Sesaminol inhibits Adipogenesis by Suppressing Mitotic Clonal Expansion and Activating the Nrf2-ARE Pathway.” The authors use a comprehensive 3T3-L1 cell culture model to investigate how Sesaminol represses adipogenesis, suggesting its potential as a promising therapeutic agent for obesity treatment. However, as is well known, the in vitro adipogenesis model differs substantially from the in vivo adipogenic process. Therefore, it would be important for the authors to provide some physiological data to support their main conclusion before this work can be considered for publication in Nutrients.

Other concerns:

  1. Quantification of the immunofluorescence staining shown in Figures 5, 6, 7, and 9 should be provided.
  2. In Figure 9, the authors conclude that Compound C represses Sesaminol-induced Nrf2 nuclear translocation. However, the images clearly show that Sesaminol plus Compound C significantly reduces overall cellular NRF2 signals compared to Sesaminol alone, suggesting that the current interpretation may not be fully accurate.
  3. The study relies solely on the 3T3-L1 cell line, which may not be sufficiently convincing. It would strengthen the findings to include additional adipocyte cell lines, and ideally experiments using human adipocytes.
  4. The AMPK–NRF2 signaling pathway regulating lipid synthesis is conserved between adipocytes and hepatocytes. Thus, it is recommended that the authors also test the function of Sesaminol in liver cells.

Round 2

Reviewer 3 Report

Comments and Suggestions for Authors

Thank you for your response to the point 2. However, I believe the current explanation can not  address the concern. The images clearly show that Sesaminol plus Compound C significantly reduces overall cellular NRF2 signals compared to Sesaminol alone. This raises the possibility that Compound C is affecting total NRF2 expression or stability, rather than specifically inhibiting nuclear translocation. To clarify this point, the authors should 1) provide quantitative data to support this conclusion including nuclear / cyto ratio and total NRF2 level; or 2) revise the interpretation of Fig 9. Without such clarification, the current interpretation remains  misleading.

Round 3

Reviewer 3 Report

Comments and Suggestions for Authors

The previous concerns have been addressed. No further issues.